

# Effects of aerobic exercise on event-related potentials related to cognitive performance: a systematic review

Julia Gusatovic, Mathias Holsey Gramkow, Steen Gregers Hasselbalch and Kristian Steen Frederiksen

Danish Dementia Research Centre, Department of Neurology, Rigshospitalet, University of Copenhagen, Copenhagen, Denmark

## ABSTRACT

**Introduction** . Aerobic exercise interventions may affect different cognitive domains such as attention, working memory, inhibition, *etc.* However, the neural mechanisms underlying this relationship, remains uncertain.

**Objective**. To perform a systematic review on exercise intervention studies that use event-related potentials (ERPs) as outcome for cognitive performance.

**Methods**. We identified studies through searches in four databases reporting the effects of either an acute bout or chronic exercise on any ERP associated with cognitive performance. Study population included participants >17 years of age with or without a diagnosis.

**Results**. A total of 5,797 records were initially identified through database searching of which 52 were eligible for inclusion. Most studies were of acute aerobic exercise with moderate intensity. Results were heterogenious across studies, but there was a trend that ERP amplitude increased and (to a lesser extent) latencies decreased post-exercise. The P3 ERP was the most often reported ERP.

**Conclusion**. Heterogeneity across studies regarding methodology limited the possibility to draw definitive conclusions but the most consistent findings were that acute aerobic exercise was associated with higher amplitudes, and to a lesser extent shorter latencies, of ERPs.

## INTRODUCTION

Aerobic exercise has been shown to improve brain health as well as cognitive functioning (*Barha et al., 2017*; *Laurin et al., 2001*; *Pope, Shue & Beck, 2003*; *Smith et al., 2010*). The physiological links between aerobic exercise and cognitive function may be facilitated through many different mechanisms, *e.g.*, secretion of neuromodulators (such as brain-derived neurotrophic factor (BDNF)), neurogenesis, increased brain plasticity and increased brain blood flow (*Waters et al., 2020*), but so far evidence for the biological mechanisms underlying this relationship remain sparse. Moreover, it may be speculated that different underlying mechanisms may mediate the effects of acute exercise interventions, which may be immediate and short-lived, versus longer exercise interventions,

Corresponding author
Kristian Steen Frederiksen,
kristian.steen.frederiksen@regionh.dk

which conversely may be slower to develop and more long-lasting. A number of studies have found an effect of long-term exercise on the hippocampus, white matter and the cortical mantle (*Colcombe et al., 2006*; *Erickson et al., 2011*; *Pajonk et al., 2010*). Although imaging studies of acute and immediate exercise effects are lacking, findings examining the effects of exercise on BDNF further indicate differential effects of acute and longer-term exercise. Specifically, BDNF was found to be elevated immediately following an acute bout of exercise, but not following 3 months of exercise (*Krogh et al., 2014*; *Tsai et al., 2021*). Nevertheless, both acute and chronic exercise seem to have beneficial effects on the brain. Acute aerobic exercise has been shown to facilitate learning mechanisms (*Perini et al., 2016*; *Voss et al., 2021*) and improve cognition in post-recovery period following exercise in healthy subjects (*Erickson et al., 2019*). A meta-analysis showed that acute bouts of aerobic exercise improved cognitive task performance (*Lambourne & Tomporowski, 2010*) and a large body of literature supports this notion (*Ratey & Loehr, 2011*). A systematic review concluded that aerobic exercise interventions exceeding one month are associated with modest improvements in attention and processing speed, executive functioning and memory (*Smith et al., 2010*). It could be theorized that chronic exercise enhances cognitive aspects by modulation of brain structure and vascular proliferation and perfusion, which develops over time, and acute exercise works by the immediate effects of neurosecretion related to exercise. However, differences in precisely which cognitive domains that are affected, and by which underlying mechanisms, by acute and chronic exercise respectively are yet to be investigated in more detail.

One method to quantify the impact of aerobic exercise on brain function and cognitive performance is by event-related potentials (ERPs), which has been widely used in studies investigating perception, attention and cognitive functioning (*Helfrich & Knight, 2019*). ERPs are small electrical potentials generated in the cortex (or subcortical generators) in response to a specific stimulus or event and can be measured noninvasively by electroencephalography (EEG) or magnetoencephalography (MEG) using scalp electrodes (*Woodman, 2010*). Information about neural activity such as early sensory perception processes and higher-level processing such as attention, cortical inhibition, response selection, error monitoring, memory update, and other cognitive functions (*Duncan et al., 2009*; *Polich, 2007*) can be obtained as different ERP components vary according to stimulus type and type of cognitive task. The most studied endogenous ERP is the P3 (300–500 ms post-stimulus), which is interpreted as an index of ability to sustain attention to a target. P3 is difficult to localize and most studies agree that P3 (P3b) has multiple dipole sources, *e.g.*, the hippocampus and the parahippocampal areas, the insula, the temporal lobe, the occipital cortex and the thalamus (*Sokhadze et al., 2017*). Another frequently studied ERP is the N2, associated with categorization, perceptual closure, inhibitory control and attention focusing (*Sokhadze et al., 2017*) and is generated by frontal and anterior cingulate cortex (*Heil et al., 2000*). Due to the noninvasiveness, ease of use and temporal resolution of the technique and the fact that it can be applied immediately after an intervention, ERPs represent an attractive method of capturing neural effects of acute bouts of exercise (*Pedroso et al., 2017*). ERPs are in general considered to express different components of executive functions, such as processing time and the amount of cognitive

resources allocated to the perception and processing of an event/task. Especially P3 seems significantly impacted by exercise in most studies.

The objective of the present study was to carry out a systematic review of studies reporting on the effects of both acute and chronic exercise on ERPs related to cognitive performance and associated behavioral measures such as accuracy and reaction time. Furthermore, we aimed to investigate whether exercise intensity was correlated with changes in ERPs.

## METHODS

### Study design and protocol registration
This study is a systematic review. Results were reported in accordance with the guidelines provided by the Preferred Reporting Items in Systematic Reviews (PRISMA) statement.

A protocol for the systematic review was registered on 09/11/2020 in the PROSPERO database (PROSPERO ID: 218808) (https://www.crd.york.ac.uk/prospero/).

### Participants, intervention, comparators
We included single group, parallel group and cross-over studies with both randomized and non-randomized allocation involving participants older than 17 years, with no history of epilepsy and sleep disorders. Interventions could be either acute aerobic exercise (single bout) or chronic aerobic exercise (>2-weeks of duration). Interventions were divided according to their exercise intensity, which for light-intensity exercise were set at <50% of $HR_{max}$, moderate-intensity at 50%–80% of $HR_{max}$ and high-intensity at >80% of $HR_{max}$. If data was not available in the studies corresponding authors were contacted, and if not possible the exercise intensity level reported was adopted. The outcome of interest was difference in ERP parameters such as latency or amplitude obtained by either EEG or MEG pre- to post intervention or between intervention group and control group pre- to post intervention. Data on effects on behavioral measures (reaction time and accuracy) were also extracted. Any paradigm judged by the authors to elicit a cognitive performance response (based on the literature on the subject) was accepted. ERPs related to processing of emotional stimuli (regardless of whether there was a cognitive element) were not included. The stimuli evoking the ERPs trials could be in any sensory modality. No limits in terms of publication year were set. Only studies in the English language and full research articles were eligible for inclusion.

### Search strategy
Searches were performed in the following databases: PubMed, Web of Science, Cochrane Library and Embase. The databases were searched from inception to the 06/NOV/2020. Mesh-terms and keywords (from the literature and thesauruses, including "exercise", "evoked potential", "event-related potential", "EEG", "electroencephalography", "MEG", "magnetoencephalography") were searched for in each database.

### Study selection, data extraction and data items
Three authors screened and selected the included studies (JG, MG, KF). Authors were blinded with regards to the results of each authors' screening results. The authors initially

screened articles on title and abstract level. Subsequently, full text articles for those identified in the first step were retrieved and assessed for final inclusion. Any disagreement with regards to whether a study could be included was resolved by consensus (JG and KF) (see Fig. 1 for flow-chart and supplementary table for more detailed explanations on exclusion reasons). Relevant data was extracted by the same author (JG) and reviewed by another author (MG). Data was extracted in an Excel data extraction sheet that was piloted using four studies before being applied to the rest of the studies. The following items were extracted: number of participants, gender, age, diagnosis, study design, characteristics of comparator and intervention (type, duration, exercise composition, intensity and how it was measured), methods (EEG/MEG used, cognitive paradigm and sensory modality, ERP outcome measurements), reported effect of the intervention on ERP measurements and behavioral results. A risk-of-bias assessment was carried out using Cochrane's Risk of Bias version 1.

## Synthesis of results

Due to large heterogeneity and according to the protocol, a qualitative synthesis of results was carried out.

## RESULTS

### Included studies

The study selection process is outlined in Fig. 1. A total of 5,797 articles were identified through database searching. Fifty-two articles representing 52 unique studies (unit of analysis) were eligible for inclusion.

### Characteristics of included studies

The 52 studies identified comprised a total of 1,734 participants. Forty-one studies investigated acute exercise interventions with the following characteristics: sample size range was 7–72, the vast majority (37 studies) included participants with a mean age range of 18–40 years, and most (38 studies) included healthy subjects only. In terms of exercise intensity, 25 studies examined the effects of moderate-intensity exercise, five examined high-intensity exercise and seven studied a combination of low-, moderate- and high-intensity exercise. The intensity could not be established in four studies. Exercise durations were primarily single bouts lasting 9–40 min and most studies used either treadmill running or exercising on a stationary bicycle. In the control conditions participants were engaged in non-exercise related activities as resting or magazine reading.

Eleven studies investigated chronic exercise interventions comprising a total of 625 participants. Sample size range was 28–141. In five studies the mean age range was 18–40, and in five studies mean age was over 60 years. One study reported a participants' age range of 40–60 years. Three studies were in healthy subjects and eight studies investigated different patient populations. Six studies investigated moderate intensity exercise, one study investigated low and high intensities and two studies examined moderate and high intensities. Two studies were of unknown exercise intensities. The duration and frequency of the exercise interventions ranged from 30–60 min sessions three times per week for three weeks to 30–60 min sessions three times per week for 24 weeks. Most studies used

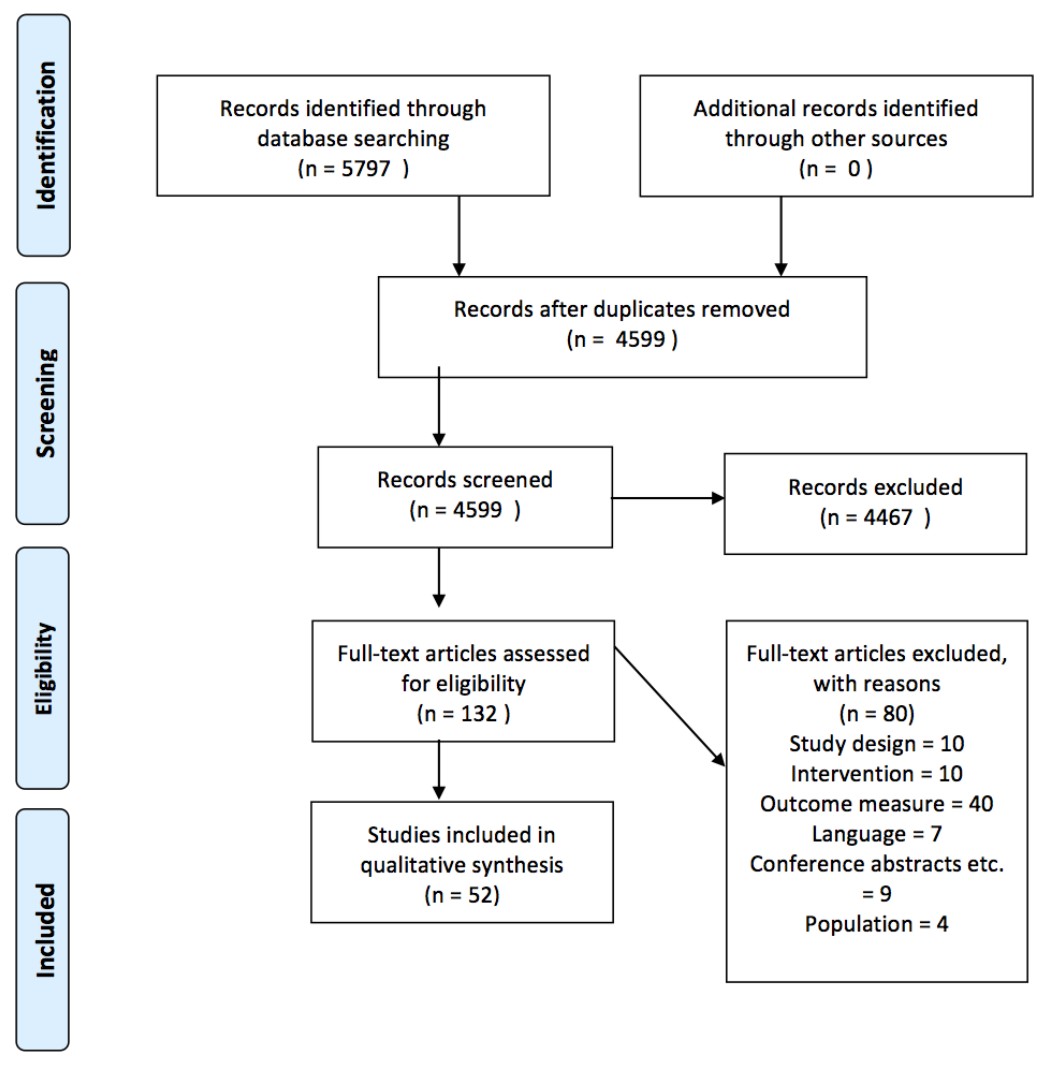

**Figure 1  Prisma flow diagram.**

a combination of running/jogging and stationary bicycling. Control conditions were non-exercise related activities (*e.g.*, usual care). All studies used EEG for the ERP assessments. See Tables 1 and 2 for study characteristics for each study included.

## Acute exercise intervention results

Due to the large heterogeneity across studies in terms of the paradigms used, ERPs investigated and methods employed, general trends in the data are difficult to extract. A total of 21 different ERPs were examined and seventeen different cognitive paradigms to elicit ERPs were used with flanker task as the most often used paradigm (12 studies). The most frequently reported ERP was the P3 (33 studies) followed by N2 and N140. Regarding P3, each study sometimes reported results for P3 amplitudes and latencies from more than one experiment as several paradigms were used to elicit the P3 or several exercise intensities were investigated, and thus numbers of experiments are reported in

**Table 1** Study characteristics (acute aerobic exercise).

| References | Design | N | Gender (% male) | Age | Diagnosis | Comparator(s) | Intervention(s) | Comments |
|---|---|---|---|---|---|---|---|---|
| Xie et al. (2020) | Counterbalanced within-subject | 16 | 100 | 24.5 ± 5.09 | Healthy | Sedentary | Ergometer cycling exercise test (single bout 30 min) | |
| Tsai et al. (2014) | Parallel group | 60 | 100 | EI(H): 22.20 ± 2.17. EI(L): 23.10 ± 2.20. NEI: 22.20 ± 1.70 | Healthy | Rest and magazine reading | Motordriving treadmilling (single bout 33 min) | |
| Yagi et al. (1999) | Single group | 24 | 50 | 20 ± 2 | Healthy | Baseline | Ergometer cycling (single bout 10 min) | |
| Walsh et al. (2019) | Counter-balanced within-subject | 25 | 36 | 22.4 ± 3.5 | Healthy | Nature documentary watching and rest | Body-weight exercises (single bout 11 min) | Exercises included burpees, squat jumps and other aerobic components |
| Scudder et al. (2012) | Counter-balanced within-subject | 37 | 51 | 19.7 ± 1.3 | Healthy | Paper reading | Motor-driven treadmilling (single bout 30 min) | |
| Swatridge et al. (2017) | Counter-balanced within-subject | 9 | 67 | 57.8 ± 11.4 | Chronic stroke | Rest | Semirecumbent stepper (single bout 20 min) | |
| Chacko et al. (2020) | Counter-balanced within-subject | 15 | 53 | 26.8 ± 5.1 | Healthy | Internet browsing | Ergometer cycling (single bout 40 min) | |
| Akatsuka et al. (2015) | Counterbalanced within-subject | 10 | 100 | 19.8 (SD not stated) | Healthy | Rest | Treadmill running (single bout 15 min) | |
| Kamijo et al. (2009) | Counter-balanced within-subject | 24 | 100 | Older: 65,5 ± 1,5. Younger: 21.8 ± 0.6 | Healthy | Baseline | 1. Light intensity ergometer cycling 2. Moderate intensity ergometer cycling (single bouts 25 min) | The participans were divided in two groups according to age (young vs old) |
| Kao, Wang & Hillman (2020) | Counter-balanced within-subject | 23 | 48 | 19.2 ± 0.6 | Healthy | Rest | Treadmilling (single bout 20 min) | |
| Aly & Kojima (2020) | Parallel group | 40 | 70 | CG: 23.10 ± 2.20. IG: 22.90 ± 2.40 | Healthy | Inactive resting | Ergometer cycling (single bout 20 min) | |

**Table 1** (*continued*)

| References | Design | N | Gender (% male) | Age | Diagnosis | Comparator(s) | Intervention(s) | Comments |
|---|---|---|---|---|---|---|---|---|
| Tsai et al. (2018) | Parallel group | 66 | 42 | AE: 65.48 ± 7.53. RE: 66.05 ± 6.64. control: 64.50 ± 6.95 | Mild cognitive impair-ment | 1. Rest and magazine reading 2. Resistance exercise | Ergometer cycling (single bout 40 min) | |
| Wollseiffen et al. (2016) | Single group | 11 | 45 | 36.5 ± 7 | Healthy | Baseline | Running (6 h) | The participants were specifically trained and experienced in ultra-marathon running |
| Wen & Tsai (2020) | Parallel group | 32 | 0 | IG: 33,13 ± 6,27 CG: 32.92 ± 7.17 | Healthy | Sitting quietly | combination of aerobic dancing and resistance training (single bout 40 min) | The study was performed in obese women |
| Chu et al. (2015) | Counter-balanced within-subject | 21 | 90 | 21.50 ± 4.68 | Healthy | Sedentary (reading) | Treadmilling (single bout 30 min) | |
| Shibasaki et al. (2019) | Single group | 15 | 100 | 20.8 ± 0.9 | Healthy | Baseline | interval cycle exercise on ergometer bicycle (four bouts of 15 min) | |
| Milankov et al. (2012) | Single group | 10 | 0 | 22.4 average, range 19–24 | Healthy | Baseline | Ergometer interval cycling (three bouts of 10 min) | |
| Ligeza et al. (2018) | Counter-balanced within-subject | 18 | 100 | 24.9 ± 2.2 | Healthy | Sitting and reading sports-related magazines | 1. Ergometer cycling moderate intensity continous 2. Ergometer cycling interval high intensity (single bouts 24 min) | |
| Pontifex et al. (2015) | Counter-balanced within-subject | 34 | 47 | 19.3 ± 0.9 | Healthy | Restful sitting | Treadmilling (single bout 20 min) | |
| Kamijo et al. (2007) | Counter-balanced within-subject | 12 | 100 | 25,7 ± 0,7 | Healthy | Baseline | 1. Ergometer cycling mild intensity 2. Ergometer cycling moderate intensity 3. Ergometer cycling hard intensity (22 min single bouts) | |

Gusatovic et al. (2022), *PeerJ*, DOI 10.7717/peerj.13604

**Table 1** (*continued*)

| References | Design | N | Gender (% male) | Age | Diagnosis | Comparator(s) | Intervention(s) | Comments |
|---|---|---|---|---|---|---|---|---|
| *Wang et al. (2020)* | Parallel group | 60 | 100 | exercise: 32.73 ± 7.15 control: 32.40 ± 7.76 | Heroin addiction | Resting and reading about heroin addiction treatments | Stationary cycle exercise (single bout 30 min) | |
| *Chu et al. (2017)* | Counter-balanced within-subject | 20 | 90 | 20.42 ± 1.16 | Healthy | Reading | Treadmilling (single bout 30 min) | |
| *Zhou & Qin (2019)* | Parallel group | 72 | 50 | 20.07 ± 0.15 | Healthy | Resting and reading | Cycling pedaling (single bout 25 min) | |
| *Rietz et al. (2019)* | Counter-balanced within-subject | 26 | 100 | 21.5 ± 2.52 | Healthy | Sitting and nature documentary watching | Ergometer cycling (single bout 30 min) | |
| *Themanson & Hillman (2006)* | Counter-balanced within-subject | 28 | 50 | higher fit: 20.1 ± 1.7. Lower fit: 20.6 ± 2.4 | Healthy | Resting and reading | Treadmilling (single bout 30 min) | |
| *Dimitrova et al. (2017)* | Parallel group | 56 | 54 | younger: 23.2 ± 2.7. Older: 70.7 ± 5.4 | Healthy | Baseline | 1. Cybercycle riding (normal exercise) 2. Cybercycle riding (exergaming) | |
| *Kao et al. (2017)* | Counter-balanced within-subject | 64 | 42 | 19.2 ± 0.8 | Healthy | Seated rest | 1. Continous aerobic exercise treadmilling (single bout 20 min) 2. High-intensity interval training (single bout 9 min) | |
| *Won et al. (2017)* | Counter-balanced within-subject | 12 | 100 | 24.8 ± 2 | Healthy | Seated rest | 1. Treadmilling (single bout 20 min) 2. Futsal (single bout 30 min) | |
| *Bae & Masaki (2019)* | Counter-balanced within-subject | 29 | 48 | 21.4 ± 1.2 | Healthy | Quiet resting | Treadmilling (single bout 30 min) | |
| *Hwang et al. (2019)* | Single group | 30 | 0 | 20.4 range 18–22 | Healthy | Baseline | Treadmilling (single bout 20 min) | |
| *Magnié et al. (2000)* | Single group | 20 | 100 | High fit group: 21.2. Low fit group: 22.9. No SD | Healthy | Baseline | Exercise protocol on a bicycle (until volitional exhaustion was reached) | |

Peer J

**Table 1** (*continued*)

| References | Design | N | Gender (% male) | Age | Diagnosis | Comparator(s) | Intervention(s) | Comments |
|---|---|---|---|---|---|---|---|---|
| *Drapsin et al. (2012)* | Single group | 24 | 0 | Judo players: 20.61 ± 3.09. Healthy: 21.06 ± 4.09 | Healthy | Baseline | 1. Ergometer cycling 60% HR max 2. ergometer cycling 75% HR max 3. ergometer cycling 90% HR max (single bouts 10 min) | |
| *Kao et al. (2018)* | Counter-balanced within-subject | 36 | 50 | 21.5 ± 3.1 | Healthy | Seated rest | 1. Treadmilling high intensity 2. Treadmilling moderate intensity (single bouts 20 min) | |
| *Wu et al. (2019)* | Counter-balanced within-subject | 30 | 57 | 21.17 ± 1.32 | Healthy | 1. resistance exercise 2. reading | Cycle ergometry (single bout 30 min) | |
| *Jain et al. (2014)* | Counter-balanced within-subject | 12 | 100 | between 18 and 21 | Healthy | Seated rest | Treadmilling (single bout, terminated on achieving any of three criteria) | (i) Volitional exhaustion, (ii) HR within 10 bpm of age predicted maximum or (iii) Rating of perceived exertion of ≥17 onBorg's Scale |
| *Winneke et al. (2019)* | Counter-balanced within-subject | 11 | 36 | 25.64 ± 3.78 | Healthy | Rest | Stationary bicycling (single bout 20 min) | |
| *Nakamura et al. (1999)* | Single group | 7 | 100 | 34.6 ± 4.7 | Healthy | Baseline | Jogging (single bout 30 min) | |
| *Yagi et al. (1998)* | Single group | 10 | 50 | mean 20.6 (no SD) | Healthy | Baseline | Ergometer cycling (single bout 10 min) | |
| *Takuro, Nishihira & Soung-Ryol (2009)* | Counter-balanced within-subject | 14 | 100 | 24.2 ± 1.3 | Healthy | NR | Stationary cycling (single bout 30 min) | |
| *Chang et al. (2017)* | Counter-balanced within-subject | 30 | 57 | 22.67 ± 1.52 | Healthy | Sedentary reading | Ergometer cycling (single bout 30 min) | |
| *Chang et al. (2015)* | Parallel group | 30 | 53 | EG: 21.67 ± 3.77. CG: 20.17 ± 1.53 | Healthy | Sedentary reading | Spinning wheel exercise | The participants were highly fit amateur basketball players |

**Notes.**

Shows characteristics of studies using acute aerobic exercise interventions.

EI(H), exercise intervention (high-intensity); EI(L), exercise intervention (low-intensity); NEI, No exercise intervention; CG, control group; IG, intervention group; NR, not registered.

Gusatovic et al. (2022), *PeerJ*, DOI 10.7717/peerj.13604

**Table 2 Study characteristics (chronic aerobic exercise).**

| References | Design | N | Gender (% male) | Age | Diagnosis | Comparator(s) | Intervention(s) | Comments |
|---|---|---|---|---|---|---|---|---|
| *Wang et al. (2017)* | Parallel group | 50 | 88 | IG: 32.3 ± 6.97 CG: 34.76 ± 7.96 | Methamphetamine dependency | Usual care | Aerobic exercise i.e., Cycling, jogging, or jump rope (30 min × 3/week in 12 weeks) | |
| *Pedroso et al. (2018)* | Parallel group | 50 | FE: 40 SG: 31 HC: 36 | FE and SG group: 78.0 ± 5.6 CG: 74.6 ± 5,3 | Alzheimers disease | 1. healthy control group 2. Social gathering (AD patients) | Exercises that stimulates aerobic endurance, flexibility, muscular resistance, and balance (60 min × 3/week in 12 weeks) | |
| *Tsai et al. (2017)* | Parallel group | 64 | 100 | O-ex: 66.88 ± 4.74. C-ex: 66.15 ± 4.90. Control: 65.70 ± 3.54 | Healthy | 1. balance and stretching (control) 2. Table tennis (open-skill exercise) | Bikeriding or brisk walking/jogging (closed-skill) (40 min × 3/week in twenty-four weeks) | |
| *Olson et al. (2017)* | Parallel group | 30 | 20 | 21.1 ± 2.0 | Major depressive disorder | Light-intensity stretching | Treadmilling og ergometer cycling (45 min × 3/week in eight weeks) | |
| *Overath et al. (2014)* | Single group | 28 | 17 | 43.3 ± 9.7 | Migraine | Baseline | Aerobic endurance programme: walking or interval jogging (40 min × 3/week in ten weeks) | |
| *Chen et al. (2020)* | Parallel group | 44 | Missing | Control: 33.87 ± 1.98. High-intensity: 32.73 ± 1.31. Moderate-intensity: 29.40 ± 1.19 | Methamphetamine-dependency | Normal daily routine | 1. Ergometer cycling (moderate-intensity) 2. Ergometer cycling (hard-intensity) (40 min × 3/week in twelve weeks) | |

Gusatovic et al. (2022), *PeerJ*, DOI 10.7717/peerj.13604

**Table 2** (*continued*)

| References | Design | N | Gender (% male) | Age | Diagnosis | Comparator(s) | Intervention(s) | Comments |
|---|---|---|---|---|---|---|---|---|
| *Zhao et al. (2020)* | Parallel group | 64 | 100 | 29.38 ± 0.56 | Metham-phetamine-dependency | Usual care | 1. Cycling on stationary bike (moderate intensity) 2. Cycling on stationary bike (high intensity) (40 min × 3/week in twelve weeks) | |
| *Özkaya et al. (2005)* | Parallel group | 44 | 68 | CG: 72.3 ± 2.1. ST: 75.8 ± 2,8. ET: 70.9 ± 3.1 | Healthy | 1. No exercise 2. Strength training | Running track (50 min × 3/week in nine weeks) | |
| *Brush et al. (2022)* | Parallel group | 55 | 32 | 20.23 ± 2.39 | Major depressive disorder | Stretching | Treadmill and ergometer cycling (45 min × 3/week in eight weeks) | |
| *Gajewski & Falkenstein (2018)* | Parallel group | 141 | 40 | 70.9 ± 5.2 | Healthy | 1. Cognitive training 2. No-contact control group 3. Social control group | Cardiovascular, aerobic, and strength exercises (90 min × 2/week in sixteen weeks) | |
| *Tsai et al. (2019)* | Parallel group | 55 | 31 | AE: 66.00 ± 7.68. RE: 65.44 ± 6.76. Control: 65.17 ± 7.00. | Mild cognitive impairment | 1. Resistance exercise 2. Static stretching exercise | Ergometer cycling and treadmilling (40 min × 3/week in sixteen weeks) | |

**Notes.**

The table shows characteristics of studies using chronic aerobic exercise interventions.

IG, intervention group; CG, control group; FE, functional exercise group; SG, social gathering; HC, healthy control; O-ex, open-skill exercise group; C-ex, closed-skill exercise group; ST, strength training; ET, endurance training; AE, aerobic exercise group; RE, resistance exercise group.

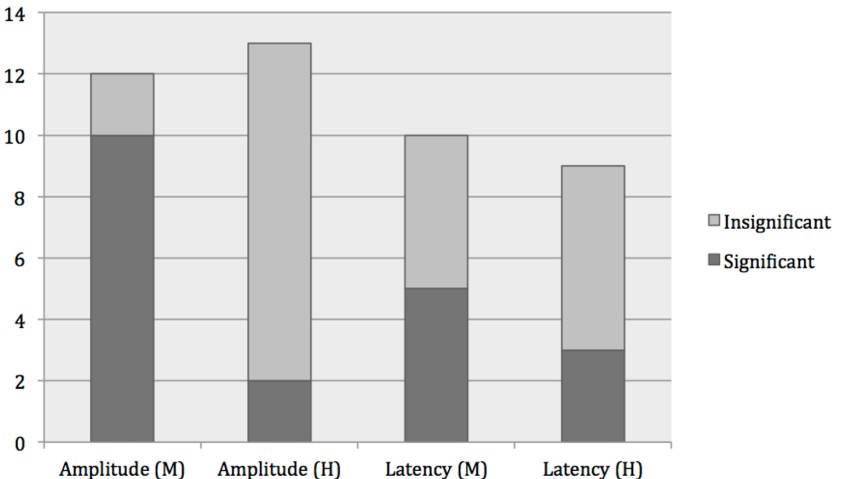

**Figure 2** **Effects of moderate-intensity exercise vs. high-intensity exercise on ERPs.** Results from studies using flanker task and acute aerobic exercise interventions, with either moderate-intensity (M) or high-intensity (H) interventions. Significant results included (1) increased amplitude after intervention in all studies, except for one study that showed a significantly decreased amplitude after intervention, and (2) decreased latencies in all studies after intervention.

the following. A total of 27 experiments out of 41 experiments reported significant effects of the exercise intervention on amplitude (25 increased, two decreased) and 16 reported effects on latency (two increased, 14 decreased). Fifteen of the 27 experiments reported significant effects on both amplitude and behavioral results (mainly decreased reaction time, only one study reported effect on accuracy (increased)) and seven experiments out of the 16 experiments reported a significant effect on latency and behavioral results (decreased reaction time). Looking across exercise intensities, there was a tendency that effects on amplitudes were significant mainly in interventions using moderate intensity exercise across all ERPs and paradigms used (see Fig. 2). Twenty-six out of 31 studies that investigated moderate-intensity interventions reported significant increases in one or more ERP component post-exercise. Conversely, out of the eleven studies that did not report any effect on ERP amplitude, two studies were of moderate intensity exercise. See Table 3 for all the results of studies using acute aerobic exercise.

## Chronic exercise results

As with the reporting of acute exercise results, results for chronic exercise studies were very heterogeneous. Nine different paradigms were used, and eight different ERPs were measured. The most frequently recorded ERPs were P3 and N2 (each recorded in six studies). Regarding P3, three studies with moderate-intensity exercise reported effect on amplitude (all increased) and concomitant effects on behavior, but none reported effect on latency. The remaining three studies reported on unknown, moderate and moderate to low-intensity exercise interventions and showed no significant effect on P3 amplitude. For N2, four studies reported effect on amplitude (all increased) and one study reported an effect on latency (decreased). All studies that reported effects on ERP

**Table 3  Acute aerobic exercise results.**

| References | Diagnosis | Recorded time after exercise | Outcome measures | Paradigm | ERP results | Behavioural results |
|---|---|---|---|---|---|---|
| **Light-intensity exercise results** | | | | | | |
| *Kamijo et al. (2009)* | Healthy | 2 min | P3 | Flanker task (visual) | Amplitude: P3↔ <br> Latency: P3↓ | Accuracy:↔ <br> Reaction time:↔ |
| *Kamijo et al. (2007)* | Healthy | 3 min | P3 | Flanker task(visual) | Amplitude: P3↑ <br> Latency: P3↓ | Accuracy:↔ <br> Reaction time:↓ |
| **Moderate-intensity exercise results** | | | | | | |
| *Tsai et al. (2014)* | Healthy | 15-20 min | P3 and CNV | Visuospatial attention task(visual) | Amplitude: P3↑ CNV↑ <br> Latency: P3↔ CNV↔ | Accuracy:↔ <br> Reaction time:↓ |
| *Yagi et al. (1999)* | Healthy | Within 10 min | P3 | Oddball task (visual and auditory) | Amplitude: P3↔ <br> Latency: P3↔ | Accuracy:↔ <br> Reaction time:↔ |
| *Scudder et al. (2012)* | Healthy | 20,2 ± 6,4min | P3 and N2 | AX-continuous performance tasks (visual) | Amplitude: P3↑ N2↔ <br> Latency: P3↔ N2↔ | Accuracy:↑ <br> Reaction time:↔ |
| *Akatsuka et al. (2015)* | Healthy | 5 min | N140 | Go-/No-go task (somato-sensory) | Amplitude: N140↑ <br> Latency: N140↔ | Accuracy:↔ <br> Reaction time:↔ |
| *Kamijo et al. (2009)* | Healthy | 2 min | P3 | Flanker task (visual) | Amplitude: P3↑ <br> Latency: P3↓ | Accuracy:↔ <br> Reaction time:↓ |
| *Kao, Wang & Hillman (2020)* | Healthy | 30 min | P3 | Serial N-back task (visual) | Amplitude: P3↑ <br> Latency: P3↔ | Accuracy:↔ <br> Reaction time:↔ |
| *Aly & Kojima (2020)* | Healthy | HR returned to within 10% of pre-ex | P2, N2c and P3 | Flanker task (visual) | Amplitude: P2↑ P3↑ N2c↑ <br> Latency: P2↔ P3↔ N2c↔ | Accuracy:↔ <br> Reaction time:↓ |
| *Wen & Tsai (2020)* | Healthy | HR returned to within 10% of pre-ex | P2, N2 and P3 | Stroop task (visual) | Amplitude: N2↓ P3↑ P2↔ <br> Latency: N2↓ P3↓ P2↔ | Accuracy:↔ <br> Reaction time:↔ |
| *Chu et al. (2015)* | Healthy | Within 10 min | P3 and N1 | Stop-signal task (visual) | Amplitude: P3↑ N1↔ <br> Latency: P3↑ N1↔ | Accuracy: NR <br> Reaction time:↓ |
| *Shibasaki et al. (2019)* | Healthy | Right after | N140 and P300 | Go-/No-go task (somatosensory) | Amplitude: N140↓ P3↔ <br> Latency: N140↔ P3↓ | Accuracy:↔ <br> Reaction time:↔ |
| *Ligeza et al. (2018)* | Healthy | HR returned to within 10% of pre-ex | N2 and P2b | Flanker task (visual) | Amplitude: P3↔ N2↑ <br> Latency: NR | Accuracy:↑ <br> Reaction time:↓ |

Gusatovic et al. (2022), *PeerJ*, DOI 10.7717/peerj-13604

**Table 3** (*continued*)

| References | Diagnosis | Recorded time after exercise | Outcome measures | Paradigm | ERP results | Behavioural results |
|---|---|---|---|---|---|---|
| *Pontifex et al. (2015)* | Healthy | HR returned to within 10% of pre-ex | P3a and P3b | Oddball task (visual) | Amplitude: P3a↔ P3b↑ <br> Latency: P3a↔ P3b↔ | Accuracy:↔ <br> Reaction time:↔ |
| *Kamijo et al. (2007)* | Healthy | 3 min | P3 | Flanker task (visual) | Amplitude: P3↑ <br> Latency: P3↓ | Accuracy:↔ <br> Reaction time:↓ |
| *Chu et al. (2017)* | Healthy | NR | P3 and conflict SP | Stroop color-word task (visual) | Amplitude: P3↑ SP↔ <br> Latency: P3↔ SP↔ | Accuracy:↔ <br> Reaction time:↓ |
| *Zhou & Qin (2019)* | Healthy | HR returned to within 10% of pre-ex | P2, N2, P3b and N450 | Stroop color-naming task (visual) | Amplitude: P2↑ N2↔ P3b↔ N450↔ <br> Latency: P2↔ N2↔ P3b↔ N450↔ | Accuracy:↔ <br><br> Reaction time:↔ |
| *Dimitrova et al. (2017)* | Healthy | Within 20 min | NR | Stroop task (visual) | Amplitude: ERP↑ <br> Latency: NR | Accuracy: n↔ <br> Reaction time:↓ |
| *Won et al. (2017)* | Healthy | HR returned to within 10% of pre-ex | P3 | Stroop color-word conflict task (visual) | Amplitude: P3↑ <br> Latency: NR | Accuracy: NR <br> Reaction time:↓ |
| *Hwang et al. (2019)* | Healthy | 90 min | N2 | Facial Go-/No-go task | Amplitude: N2↓ <br> Latency:↔ | Accuracy:↔ <br> Reaction time:↔ |
| *Kao et al. (2018)* | Healthy | NR | P3 | Flanker task (visual) | Amplitude: P3↑ <br> Latency: P3↔ | Accuracy:↔ <br> Reaction time:↓ |
| *Wu et al. (2019)* | Healthy | 30 min | P3b and N1 | Task-switching test | Amplitude: P3↑ N1↔ <br> Latency: NR | Accuracy:↔ <br> Reaction time:↓ |
| *Winneke et al. (2019)* | Healthy | 2,56 min (range 2 –3,10) | N2 and P3 | Flanker task | Amplitude: N2↑ P3↔ <br> Latency: N2↔ P3↓ | Accuracy:↔ <br> Reaction time:↔ |
| *Takuro, Nishihira & Soung-Ryol (2009)* | Healthy | Right after + when HR had returned to pre-ex values | P3, early and late CNV | Go-/No-go reaction time task | Amplitude: P3↑ early CNV↑ late CNV↑ (only right after) <br> Latency: P3↔ early CNV↔ late CNV↔ | Accuracy: NR <br><br><br> Reaction time: NR |
| *Chacko et al. (2020)* | Healthy | Right after | BP, pN, N1, pN1, pP1 and P3 | Discriminative re-sponse task | Amplitude: N1↓ all other↔ <br> Latency: pN1↓ all other↔ | Accuracy:↔ <br><br> Reaction time:↔ |
| *Milankov et al. (2012)* | Healthy | NR | P3 | Oddball task (audi-tory) | Amplitude: P3↔ <br> Latency: P3↑ | Accuracy: NR <br> Reaction time: NR |
| *Bae & Masaki (2019)* | Healthy | HR returned to within 10% of pre-ex | P3 | Task-switching paradigm (visual) | Amplitude: P3↑ <br> Latency: P3↓ | Accuracy:↔ <br> Reaction time:↓ |

**Table 3** (*continued*)

| References | Diagnosis | Recorded time after exercise | Outcome measures | Paradigm | ERP results | Behavioural results |
|---|---|---|---|---|---|---|
| *Drapsin et al. (2012)* | Healthy | Right after | P3 | Oddball task (auditory) | Amplitude: P3↑ | Accuracy:↔ |
| | | | | | Latency: P3↔ | Reaction time:↔ |
| *Swatridge et al. (2017)* | Chronic stroke | Both 0, 20 and 40 min after | P3 | Flanker task (visual) | Amplitude: P3↑ (40 min post-ex) | Accuracy:↔ |
| | | | | | Latency: P3↓ (20 min post-ex) | Reaction time:↔ |
| *Tsai et al. (2018)* | Mild cognitive impair-ment | HR returned to within 10% of pre-ex | P3 | Flanker task (visual) | Amplitude: P3↑ | Accuracy:↔ |
| | | | | | Latency: P3↔ | Reaction time:↓ |
| *Wang et al. (2020)* | Heroin addic-tion | HR returned to within 10% of pre-ex | N2 and N2d | Go-/No-go task (visual) | Amplitude: N2↑ N2d↑ | Accuracy:↑ |
| | | | | | Latency: N2↔ N2d↔ | Reaction time:↔ |
| *Chang et al. (2017)* | Healthy | 15 min after | N1, N2, P3 and N450 | Stroop task (visual) | Amplitude: P3↑ N450↑ N1↔ N2↔ | Accuracy:↔ |
| | | | | | Latency: N450↓ | Reaction time:↓ |
| *Chang et al. (2015)* | Healthy | Within in 10 minutes | P3 | Attention network task | Amplitude: P3↑ | Accuracy:↔ |
| | | | | | Latency: P3↔ | Reaction time:↑ |
| **High-intensity exercise results** | | | | | | |
| *Xie et al. (2020)* | Healthy | Within 15 min | P3 and LPP | Flanker task (visual) | Amplitude: P3↔ LPP↑ | Accuracy:↔ |
| | | | | | Latency: NR | Reaction time:↓ |
| *Walsh et al. (2019)* | Healthy | 10 min | RewP | Novel gambling task (visual) | Amplitude: RewP↓ | Accuracy:↔ |
| | | | | | Latency: NR | Reaction time:↔ |
| *Ligeza et al. (2018)* | Healthy | HR returned to within 10% of pre-ex | N2 and P2b | Flanker task (visual) | Amplitude: N2↔ P2b↔ | Accuracy:↔ |
| | | | | | Latency: NR | Reaction time:↔ |
| *Kamijo et al. (2007)* | Healthy | 3 min | P3 | Flanker task (visual) | Amplitude: P3↔ | Accuracy:↔ |
| | | | | | Latency: P3↓ | Reaction time:↓ |
| *Rietz et al. (2019)* | Healthy | 30 min | P3, CNV, N2 | Continous perfor-mance task (visual) | Amplitude: P3↑ CNV↔ N2↔ | Accuracy:↔ |
| | | | | | Latency: P3↔ CNV↔ N2↔ | Reaction time:↔ |
| *Rietz et al. (2019)* | Healthy | 41 min | P3, CNV, N2 | Flanker task (visual) | Amplitude: P3↔ CNV↔ N2↔ | Accuracy:↔ |
| | | | | | Latency: P3↔ CNV↔ N2↔ | Reaction time:↔ |
| *Rietz et al. (2019)* | Healthy | 54 min | P3, CNV, N2 | Four-choice reaction-time task (visual) | Amplitude: P3↔ CNV↔ N2↔ | Accuracy:↔ |
| | | | | | Latency: P3↔ CNV↔ N2↔ | Reaction time:↔ |
| *Kao et al. (2017)* | Healthy | 20 min | P3 | Flanker task (visual) | Amplitude: P3↓ | Accuracy:↔ |
| | | | | | Latency: P3↓ | Reaction time:↓ |

Gusatovic et al. (2022), *PeerJ*, DOI 10.7717/peerj.13604

**Table 3** (*continued*)

| References | Diagnosis | Recorded time after exercise | Outcome measures | Paradigm | ERP results | Behavioural results |
|---|---|---|---|---|---|---|
| *Kao et al. (2018)* | Healthy | NR | P3 | Flanker task (visual) | Amplitude: P3↔ <br> Latency: P3↓ | Accuracy:↔ <br> Reaction time:↓ |
| *Milankov et al. (2012)* | Healthy | NR | P3 | Oddball task (auditory) | Amplitude: P3↓ <br> Latency: P3↔ | Accuracy: NR <br> Reaction time: NR |
| *Themanson & Hillman (2006)* | Healthy | HR returned to within 10% of pre-ex | Error negativity, error positivity and N2 | Flanker task (visual) | Amplitude: all↔ <br> Latency: all↔ | Accuracy:↔ <br> Reaction time:↔ |
| *Drapsin et al. (2012)* | Healthy | Right after | P3 | Oddball task (auditory) | Amplitude: P300↔ <br> Latency: P300↔ | Accuracy:↔ <br> Reaction time:↔ |
| *Jain et al. (2014)* | Healthy | HR returned to within 10% of pre-ex | P3 | Oddball task (auditory) | Amplitude: P3↑ <br> Latency: P3↓ | Accuracy: NR <br> Reaction time: NR |
| **Unknown intensity results** | | | | | | |
| *Wollseiffen et al. (2016)* | Healthy | NR | P2 and N1 | Chalkboard challenge | Amplitude: P2↔ N1↔ <br> Latency: P2↔ N1↔ | Accuracy:↔ <br> Reaction time:↔ |
| *Magnié et al. (2000)* | Healthy | When body temperature and HR had returned to pre-exercise levels | P3, P2, N1 and N2 | Oddball task (auditive) | Amplitude: P3↑ P2↔ N1↔ N2↔ <br> Latency: P3↓ P2↔ N1↔ N2↔ | Accuracy:↔ <br><br> Reaction time:↔ |
| *Nakamura et al. (1999)* | Healthy | 10 min | P3, P2, N100 and N2 | Oddball task (auditory) | Amplitude: P3↑ P2↑ N100↔ N2↔ <br> Latency: P3↔ P2↔ N100↔ N2↔ | Accuracy:↔ <br><br> Reaction time:↔ |
| *Yagi et al. (1998)* | Healthy | Right after | P3 | Oddball task (visual) | Amplitude: P3↔ <br> Latency: P3↔ | Accuracy:↔ <br> Reaction time:↔ |

**Notes.**

Shows results for all ERPs investigated with different cognitive paradigms using acute aerobic exercise interventions. Arrows (↑) indicate increase in measure following exercise, arrows (↓) indicate decrease in measure following exercise and arrows (↔) indicate no difference.

HR, Heart rate; NR, not registered; CNV, contingent negative variation; LPP, late positive potential; RewP, reward positivity.

amplitude also reported an effect on behavioral results. Across intensity, as for acute exercise interventions, only studies of moderate-intensity exercise reported effects on ERPs. See Table 4 for all the results of studies using chronic aerobic exercise.

## DISCUSSION

The aim of the present study was to carry out a systematic review assessing the impact of aerobic exercise on ERPs related to cognitive performance. Heterogeneity across studies regarding methodology limited the possibility to draw definitive conclusions but the most consistent findings were that acute aerobic exercise was associated with higher amplitudes, and to a lesser extent shorter latencies, of ERPs. Moderate-intensity exercise was the most effective exercise intensity across studies in terms of affecting ERPs. For chronic exercise only a few studies were identified and results were less consistent. Lastly, in about half of the studies reporting an effect on ERPs, behavioral outcomes were also affected by the interventions.

Our findings are consistent with previous findings in another systematic review that evaluated the influence of physical activity or exercise on P3 in elderly participants. The authors concluded that physical activity and physical exercise positively influences changes in amplitude (*Pedroso et al., 2017*). Findings also suggested that P3 latency was less sensitive to physical activity compared to amplitude (*Pedroso et al., 2017*), which also aligns with the findings of the present systematic review.

P3 was the most frequently reported ERP in both acute and chronic exercise interventions in the identified studies. P3 represents the amount of attentional resources that is allocated to a specific task. Shorter latencies represent faster processing and higher amplitudes may be associated with attentional functioning (*Polich & Heine, 1996*). Our findings suggest that acute and to a lesser extent chronic exercise interventions seem to affect P3 amplitude. This aligns well with other studies that have found that exercise had a positive impact on attentional functioning and cognitive performance (*Northey et al., 2018*; *Radel, Tempest & Brisswalter, 2018*; *Waters et al., 2020*).

In the identified studies, other ERPs were also investigated such as the N1, N140 and N2 with the latter being the most frequently reported. Here, findings were more conflicting with studies reporting increases, decreases and no effects on amplitude and latency. These discrepancies may be due to difference in terms of intervention, control condition, paradigm used and study population, and further conclusions regarding these ERPs in relation to exercise are not possible.

We found that effects on ERPs seemed dependent on the exercise intensity, as most significant results for amplitudes were found in studies using moderate-intensity exercise. It has been proposed that P3 amplitude changes may be described by an inverted U-shaped curve relative to exercise intensity (*Kamijo et al., 2004*) and results from our study support this. It is uncertain what may mediate the U-shaped relationship indicating that low-intensity is "not enough" whereas high-intensity is "too much". Interestingly, in a meta-analysis of patients with dementia, lower-frequency exercise interventions were more effective in improving cognition than higher-frequency interventions (*Groot*

**Table 4   Chronic aerobic exercise results.**

| References | Exercise intensity | Population | Duration | ERPs reported | Paradigm | ERP results | Behavioural results |
|---|---|---|---|---|---|---|---|
| *Tsai et al. (2017)* | M | Healthy | 40 min × 3 /week in 24 weeks | P3 | Task switching (visual) | Amplitude: P3↑ | Accuracy:↔ |
| | | | | | | Latency: P3↔ | Reaction time:↓ |
| *Tsai et al. (2017)* | M | Healthy | 40 min × 3 /week in 24 weeks | P3 | N-back task (visual) | Amplitude: P3↑ | Accuracy:↑ |
| | | | | | | Latency: P3↔ | Reaction time:↔ |
| *Özkaya et al. (2005)* | M | Healthy | 50 min × 3 /week in 9 weeks | N1, P2, N2, and P3 | Oddball task (auditory) | Amplitude: all↔ | Accuracy: NR |
| | | | | | | Latency: N2↓ P2↓ N1↔ P3↔ | Reaction time: NR |
| *Gajewski & Falkenstein (2018)* | NR | Healthy | 90 min × 2 /week in 16 weeks | P3a and P3b | N-back task (visual) | Amplitude: P3a↔, P3b↔ | Accuracy:↔ |
| | | | | | | Latency: NR | Reaction time:↔ |
| *Wang et al. (2017)* | M | Metham-phetamine dependency | 30 min × 3 /week in 12 weeks | N2 | Standard Go-/No-go task (visual) | Amplitude: N2↑ | Accuracy:↑ |
| | | | | | | Latency: NR | Reaction time:↔ |
| *Wang et al. (2017)* | M | Metham-phetamine dependency | 30 min × 3 /week in 12 weeks | N2 | Methamphetamine-related Go-/No-go task (visual) | Amplitude: N2↑ | Accuracy:↑ |
| | | | | | | Latency: NR | Reaction time:↔ |
| *Pedroso et al. (2018)* | L to M | Alzheimers disease | 60 min × 3 /week in 12 weeks | P3 | Oddball task (auditory) | Amplitude: P3↔ | Accuracy: NR |
| | | | | | | Latency: P3↔ | Reaction time: NR |
| *Olson et al. (2017)* | M | Major depressive disorder | 45 min × 3/week in 8 weeks | N2 | Flanker task (visual) | Amplitude: N2↑ | Accuracy:↔ |
| | | | | | | Latency: NR | Reaction time:↓ |
| *Overath et al. (2014)* | NR | Migraine | 40 min × 3/week in ten weeks | CNV | Trail making test | Amplitude: CNV↓ | Accuracy:↔ |
| | | | | | | Latency: NR | Reaction time:↓ |
| *Overath et al. (2014)* | NR | Migraine | 40 min × 3/week in ten weeks | CNV | d2-letter cancellation test | Amplitude: CNV↓ | Accuracy:↔ |
| | | | | | | Latency: NR | Reaction time:↓ |
| *Chen et al. (2020)* | M | Metham-phetamine dependency | 40 min/week in 12 weeks | N1 and P2 | 2-back task (visual) | Amplitude: N1↔ P2↔ | Accuracy:↔ |
| | | | | | | Latency: NR | Reaction time:↔ |
| *Chen et al. (2020)* | H | Metham-phetamine dependency | 40 min × 3 /week in 12 weeks | N1 and P2 | 2-back task (visual) | Amplitude: N1↓ P2↔ | Accuracy:↑ |
| | | | | | | Latency: NR | Reaction time:↓ |
| *Zhao et al. (2020)* | M | Metham-phetaminee dependency | 40 min × 3/week in 12 weeks | N2 and P2 | Temporal discounting task (visual) | Amplitude: P2↑ N2↑ | Accuracy: NR |
| | | | | | | Latency: NR | Reaction time:↓ |
| *Zhao et al. (2020)* | H | Methamphetamine dependency | 40 min × 3/week in 12 weeks | N2 and P2 | Temporal discounting task (visual) | Amplitude: P2↔ N2↔ | Accuracy: NR |
| | | | | | | Latency: NR | Reaction time:↔ |
| *Brush et al. (2022)* | M | Major depressive disorder | 45 min × 3/week in 8 weeks | ERN | Flanker task (visual) | Amplitude: ERN↔ | Accuracy: NR |
| | | | | | | Latency: NR | Reaction time: NR |
| *Brush et al. (2022)* | M | Major depressive disorder | 45 min × 3/week in 8 weeks | RewP | Doors task (visual) | Amplitude: RewP↔ | Accuracy: NR |
| | | | | | | Latency: NR | Reaction time: NR |
| *Tsai et al. (2019)* | M | Mild cognitive impairment | 40 min × 3/week in 16 weeks | P2 and P3 | Task switching paradigm | Amplitude: P2↔ P3↑ | Accuracy:↑ |
| | | | | | | Latency: P2↔ P3↔ | Reaction time:↑ |

**Notes.**

Shows results for all ERPs investigated with different cognitive paradigms using chronic aerobic exercise interventions. Arrows (↑) indicate increase in measure following exercise, arrows (↓) indicate decrease in measure following exercise and arrows (↔) indicate no difference.

HR, Heart rate; NR, not registered; CNV, contingent negative variation; RewP, reward positivity; ERN, error-related negativity.
_et al., 2016a_) also hinting at the concept that not all doses of exercise are beneficial. In an observational study, a differential effect of exercise on cognitive functions was found, as physical activity was found to be positively associated with executive function and processing speed and negatively with memory (_Frederiksen et al., 2015_). Further studies are needed however, as results are conflicting (_Hoffmann et al., 2016_), and studies comparing different exercise intensities directly are few (_Kamijo et al., 2007_; _Kao et al., 2017_; _Wang et al., 2016_).

The use of ERPs in measuring cognitive performance post-exercise is practical and informative as the method enables assessment immediately before and after exercise. Further, temporal sensitivity is high, so ERP components can be tracked during cognitive paradigms. However, the spatial sensitivity is lacking and the different ERP components are difficult to locate (_Woodman, 2010_). Linking ERP findings with structural and functional MRI would give valuable information in this regard. However, studies are lacking in which MRI pre- and post-exercise has been carried out, especially in studies using acute exercise interventions. Future research should focus on concomitant use of MRI and ERPs in the investigation of cognitive responses, as the methods are complementary.

A number of aspects that possibly affect ERPs in relation to cognitive performance include age, exercise modality and exercise duration. We therefore divided the studies in acute and long-term aerobic exercise, although it can be theorized that smaller distinctions in duration could as well affect ERPs differently, _e.g._, an exercise session under 20 min versus over 20 min. The majority of the studies included using acute exercise interventions had exercise duration between 9–40 min and no apparent difference was observed. Age related differences are also worth taking into account when interpreting ERP results, as studies have shown a latency increase and a P3 amplitude decrease in healthy senior individuals compared to younger individuals. Healthy seniors compared to individuals with dementia show further increase in latencies and decrease in amplitudes, which had led to the suggestion that P3 could be considered as a biological marker of cognitive impairment (_Hedges et al., 2016_; _O'Mahony et al., 1996_; _Pedroso et al., 2012_). Elderly individuals' ERPs post-exercise could therefore be more susceptible to exercise than younger individuals (_Hillman et al., 2002_). Another ERP component that was frequently reported was N2, which is involved in inhibitory control. An increase in N2 amplitude is found to correlate with correctly inhibited no-go stimuli in both younger and older adults, whereas P3 amplitude in the same study showed an age-related decrease (_Kardos, Kóbor & Molnár, 2020_). This adds several variables to the issue, where it seems that some ERP components are age-dependent and some are not, but included studies in the systematic review investigating N2 were few and we refrain from concluding anything based on these results.

Several limitations were present in the studies included in the systematic review. There were concerns regarding risk of bias in all studies included. In study designs using aerobic exercise interventions it is almost impossible to blind participants, as aerobic exercise is not possible to mask and therefore performance bias is a risk. Furthermore, the studies were also difficult to compare, as ERPs were elicited through different sensory modalities by various cognitive tests that examined different aspects of cognitive performance. The

investigators most often also examined different ERPs, such as P3 or N2, making direct comparisons between studies difficult.

The systematic review also has limitations. We chose to include studies on populations that were both healthy and with different diseases in our synthesis, which could have biased our results. Studies reporting on diseased participants were few and in general no discernable trends in the findings convincingly indicated a different response between healthy participants and diseased. We chose to focus on aerobic exercise interventions as these have shown more robust and consistent effects on cognition (*Groot et al., 2016*). However, other exercise types such as resistance training may have similar effects and therefore it cannot be ruled out that an effect on ERPs is also present for these types of exercise. We had very broad inclusion criteria in terms of paradigms and ERPs and having instead focused on one or two paradigms and ERPs would have perhaps left less uncertainty in terms of interpretation and may have enabled a meta-analysis. However, by including as many paradigms and ERPs as possible, we will enable researchers in having an overview of those used in exercise research. Thus we here present a wider review compared to the previous systematic review both in terms of population, intervention and ERPs examined (*Pedroso et al., 2017*).

In conclusion, we found that aerobic exercise, especially acute exercise, affected amplitudes and also to a lesser extent latencies of ERP components. Most studies focused on acute aerobic exercise in healthy participants and future research should focus more on (1) which role acute versus chronic exercise play in regards to ERP amplitudes and latencies, and (2) whether ERP amplitudes and latencies are dependent on exercise intensity. Future studies should focus on comparing these aspects of aerobic exercise directly.

### Funding
The authors received no funding for this work.

### Competing Interests
The authors declare there are no competing interests.

### Author Contributions
- Julia Gusatovic conceived and designed the experiments, performed the experiments, analyzed the data, prepared figures and/or tables, authored or reviewed drafts of the article, and approved the final draft.
- Mathias Holsey Gramkow analyzed the data, authored or reviewed drafts of the article, and approved the final draft.
- Steen Gregers Hasselbalch conceived and designed the experiments, authored or reviewed drafts of the article, and approved the final draft.
- Kristian Steen Frederiksen conceived and designed the experiments, analyzed the data, authored or reviewed drafts of the article, and approved the final draft.

## Data Availability

The extraction sheet with all data on the studies included is available in the Supplementary File.

## Supplemental Information

Supplemental information for this article can be found online at http://dx.doi.org/10.7717/peerj.13604#supplemental-information.

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
