# Peer review of "Effects of aerobic exercise on event-related potentials related to cognitive performance: a systematic review"

_PeerJ, doi:10.7717/peerj.13604_

## Round 0.1 · original submission · Minor Revisions

I am writing to inform you that your manuscript it requires a number of Minor Revisions.

We are waiting the requested changes.

Kind regards,

Badicu Georgian, Ph.D
Academic Editor, PeerJ

·

Basic reporting

The paper is generally written well and easy to read. The topic is interesting and important. Sufficient background was provided in the introduction. Tables and figures are clear. There are some places that should be improved.
1. It would be good to include discussion on how ERP paradigms, features of study populations and dosage and lengths would affect the results. The authors have briefly mentioned some of these either in main text or table, it might be worth the address a little bit in discussion and consider how these would inform future research.
2. "Future studies should focus on comparing these aspects of aerobic exercise head-to-head". I am not sure what this means "head to head". Could you clarify.
3. As EEG is not very sensitive to structure change, it might be worth to link the current ERP findings with recent MRI findings in the discussion and see if there are consistent trend.
4. There are similar systematic review on this topic which mentioned in the introduction. Should the authors highlight what additional knowledge has bee added by this review.

Experimental design

See my comments above.

Validity of the findings

See above

Reviewer 2 ·

Basic reporting

The review presents a systematic review of exercise intervention studies on event-related potentials. The rationale for the review is well justified as there are not enough integrative views on this rapidly developing field. The manuscript is well written, and the review process seems to be adequate.

My overall perception of the work is positive, although some issues limit my enthusiasm for this manuscript in its present form. These are relatively minor comments, and my with presenting them is not to be critical but to help increase the impact of the review.

Background / context
1). The title, abstract, and to some extent, introduction suggest that the review concerns the effects of aerobic exercise on ERPs. However, the review focuses on ERPs in response to tasks that measure cognitive performance. So the title should be more specific.

In the method session, the authors clarify that they focus on ERPs associated with cognitive function, but I think that this is also too general. In my opinion, the presented review does not focus on cognitive function but on cognitive performance. For example, some studies rejected from the review concern the processing of emotional stimuli (e.g., Brush et al., Ligeza et al.). These studies used ERP / EMF as markers of attention towards emotional stimuli. Attention is also a cognitive function. I would suggest that authors consider framing this review around a more specific topic: the effects of aerobic exercise on ERPs associated with cognitive performance, not with ERPs associated with cognitive functions, and certainly not an impact of exercise on ERPs in general.

2) I think that providing more rationale for concentrating on aerobic, but not other (e.g. resistance) forms of training could be useful.

3) The N2 component is often considered a marker of the inhibition process. I think it is worth adding this function while reviewing and discussing the results and the functional significance of the component.

Experimental design

4) The authors divided interventions according to their exercise intensity based on % of HRmax However, not all the studies considered based intensity on % HRmax. How did the authors conclude about the intensity of exercise in these studies, when the intensity was based on a subjective (RPE) scale, ventilatory thresholds, or % HRR (heart rate reserve)?

5) I do not understand the reasons for excluding the following two Chang's studies (https://doi.org/10.1111/psyp.12784; https://doi.org/10.3389/fnhum.2015.001560. The authors provide "the wrong outcome measure" as a reason, but the measure of this study was the cognitive performance in typical tasks measuring cognitive functions with a combined behavioral and ERP measurement.

Validity of the findings

6) I condemn that the authors distinguish between acute and chronic exercise effects. I think they may have discussed a little more about the possible mechanisms behind acute and chronic effects, as they seem to be different.

7) Relatively, I think it would be worth trying to make conclusions about the effects of different durations (e.g under 20 minutes vs over 20minutes) or modalities of exercise (e.g., running vs. cycling).

---

## Round 0.2 · accepted · Accept

Thank you for submitting the manuscript to PeerJ. Great improvements were performed in the manuscript. Currently, the article is acceptable for publication.

We look forward to hearing from you soon.

Best wishes,

Badicu Georgian, Ph.D

Reviewer 2 ·

Basic reporting

The authors addressed all my previous remarks very carefully. It only remains for me to congratulate the authors on an exciting review!

Experimental design

no comment

Validity of the findings

no comment